# CircTCF4 Suppresses Proliferation and Differentiation of Goat Skeletal Muscle Satellite Cells Independent from AGO2 Binding

**DOI:** 10.3390/ijms232112868

**Published:** 2022-10-25

**Authors:** Shuailong Zheng, Li Li, Helin Zhou, Xujia Zhang, Xiaoli Xu, Dinghui Dai, Siyuan Zhan, Jiaxue Cao, Jiazhong Guo, Tao Zhong, Linjie Wang, Hongping Zhang

**Affiliations:** Farm Animal Genetic Resources Exploration and Innovation Key Laboratory of Sichuan Province, College of Animal Science and Technology, Sichuan Agricultural University, Chengdu 611130, China

**Keywords:** circRNA, skeletal muscle satellite cells, proliferation, differentiation

## Abstract

The proliferation and differentiation of mammalian skeletal muscle satellite cells (MuSCs) are highly complicated. Apart from the regulatory signaling cascade driven by the protein-coding genes, non-coding RNAs such as microRNAs (miRNA) and circular RNAs (circRNAs) play essential roles in this biological process. However, circRNA functions in MuSCs proliferation and differentiation remain largely to be elucidated. Here, we screened for an exonic circTCF4 based on our previous RNA-Seq data, specifically expressed during the development of the longest dorsal muscle in goats. Subsequently, the circular structure and whole sequence of circTCF4 were verified using Sanger sequencing. Besides, circTCF4 was spatiotemporally expressed in multiple tissues from goats but strikingly enriched in muscles. Furthermore, circTCF4 suppressed MuSCs proliferation and differentiation, independent of AGO2 binding. Finally, we conducted Poly(A) RNA-Seq using cells treated with small interfering RNA targeting circTCF4 and found that circTCF4 would affect multiple signaling pathways, including the insulin signaling pathway and AMPK signaling pathway related to muscle differentiation. Our results provide additional solid evidence for circRNA regulating skeletal muscle formation.

## 1. Introduction

Mammalian skeletal muscle growth and development is a complex physiological process involving multiple transcription factors and signaling pathways co-regulation through spatiotemporal expression patterns [1,2,3]. These transcription factors primarily include the myogenic regulatory factor family, such as the *Pax* gene family and the myogenic enhancement factor family. Myogenesis is also regulated by Wnt, BMP4, Sonic Hedgehog, and P38 MAPK signaling pathways, participating in the growth and development of skeletal muscle satellite cells (MuSC) [4,5,6,7,8,9].

The maintenance and repair of muscle depend on the MuSC, a muscle-derived stem cell with superior proliferation and differentiation potential [10]. Alexander Mauro first discovered that MuSCs locate plasma membranes adjacent to muscle fibers under the muscle fiber substrate [9]. Subsequent studies have shown that MuSCs are resting under steady-state conditions. Still, when the skeletal muscle is subjected to physical trauma or induced stimulation, MuSCs are immediately activated [11]. After activation, the number of satellite cells is continuously expanded through symmetric division, and a settled satellite cell population is generated in the asymmetric division [12]. Finally, new muscle fibers are formed through proliferation and differentiation to repair the damaged muscle [13].

Circular RNA (CircRNA), generated from mRNA precursors with unique reverse back-splicing mode [14,15], is a kind of RNA molecule covalently linked end to end without 5′ Caps and a 3′ poly (A) structure. According to the exon and intron composition, circRNAs are generally classified into three types: intron circular RNAs (ciRNAs), exon-intron circRNAs (EIciRNAs), and exon circRNAs (EcircRNAs) [16,17,18]. CircRNA is generally characterized by structural stability, conserved sequence, and temporal, tissue, or cell-type specificity [19,20,21,22]. Moreover, CircRNA regulates genes at transcriptional, post-transcriptional, and even translational levels through varied mechanisms [23], including sponging miRNA, enriching RNA-binding proteins, or translating peptides [24,25,26].

*TCF4* is a member of the basic/helix-loop-helix (bHLH) transcription factor family [27]. These proteins are involved in various developmental processes, including controlling proliferation and determining cell fate [28], as transcriptional integrators of adaptive cellular processes in terminally differentiated cells [29]. Using high-throughput data, *TCF4* was reported to be transcribed into circular isoforms (circTCF4) in a cell-type-specific manner [30]. For example, circTCF4 was more enriched in Hela cells than in human lung cancer cells (A549). In addition, circTCF4 is highly expressed in the brain and synapses, indicating a potential function in regulating neural development [30]. Although TCF4 dominantly derivates more circTCF4 than linear transcripts [30], the myogenic function of circTCF4 and underpinning molecular mechanisms remains unveiled [31,32].

Based on the circular RNA-specific transcriptome sequencing results of the goat longissimus dorsi muscle at different stages, we screened for circTCF4, a circRNA molecule with spatial and temporal expression specificity. circTCF4 was consistently highly expressed at 105 d (when the muscle fibers started to form) and 3 d and 60 d after birth, suggesting that it may be involved in regulating muscle growth and development [33,34]. Combined with the results of previous experiments, we selected circTCF4 as a candidate circRNA, and focused on the core question of whether circTCF4 regulates myoblast proliferation and differentiation in this experiment. Our results showed that circTCF4 can efficiently inhibit the proliferation and differentiation of MuSCs. Further examination revealed that *circTCF4* functions independently from AGO2 binding. These will shed light on the current understanding of circRNA function during muscle progression.

## 2. Results

### 2.1. Characterization of Goat CircTCF4 Sequence

Based on the unique sequence of circTCF4 obtained from high-throughput circRNA-seq data, using RNA derived from newborn goat LD muscle and convergent primers, we amplified and obtained goat circTCF4, a 428 nt transcript generated by exons 11–14 of TCF4 gene (Gene ID: 102188726). The putative circTCF4 junction was verified using Sanger sequencing (Figure 1A), consistent with the high-throughput data.

Moreover, to confirm the circularity of circTCF4, we designed a pair of divergent primers and then compared the amplification results using several cDNA templates that differed in reverse-transcript primers. The amplicons of linear GAPDH transcripts successfully appeared in cDNA templates reverse-transcripted with Random or OligdT primers and genome DNA (gDNA), circTCF4 amplicon only presented in the Random primer reverse-transcripted template (Figure 1B).

Since circular RNAs were more stable than linear transcripts, we treated RNA samples with RNase R and then quantified circTCF4 levels via qPCR. As expected, circTCF4 decreased slightly by RNase R treatment; meanwhile, it caused almost complete degradation of GAPDH mRNAs (Figure 1B,C). These results indicate that circTCF4 is an end-to-end covalently linked circRNA.

### 2.2. CircTCF4 Is Enriched in Developmental Skeletal Muscles and MuSCs

To systematically profile the spatiotemporal pattern of circTCF4 in tissues, using RNAs extracted from LD muscle of goats aged from E45 (embryo 45 d) to 150 days postnatal (B150), we found that the level of circTCF4 gradually increased during gestation, peaked at newborn, and thereafter downregulated (Figure 2B). Moreover, using five different muscles (LD muscle, psoas major, semimembranosus, semitendinosus, and gastrocnemius), cerebrum (part of the brain), and five internal organs (heart, lung, kidney, spleen, liver) from 3-days goats, we further anchored the enrichment of circTCF4 in muscles, including the heart (Figure 2A). These results suggest that circTCF4 plays an essential role in muscle development.

Furthermore, we detected the expression of circular (circTCF4) and linear transcripts (TCF4 mRNA) originating from the TCF4 gene during the proliferation and differentiation of MuSCs. We found both were low at the proliferation stage but gradually increased once cells shifted to differentiation and increased dramatically after that. In addition, circTCF4 and TCF4 mRNA expression patterns were similar, with Spearman correlation coefficient ρ = 0.95, implying they were closely related (Figure 2C).

### 2.3. CircTCF4 Retained Goat MuSCs Proliferation

To verify the effect of circTCF4 on the growth of myoblasts, we firstly transfected cells with the small interfering RNA targeting circTCF4 (si-circTCF4). Compared with the control (siNC), circTCF4 levels were almost halved by si-circTCF4 treatment (*p* < 0.01) (Figure 3A), accompanied by the insignificantly varied expression of TCF4 mRNA (*p* > 0.05) (Figure 3B). Noteworthy, the expression of Pax7, a MuSCs’ proliferation-related marker gene, increased significantly after interfering with circTCF4; in contrast, expression of PCNA mRNA, CCND1 mRNA, CDK2 mRNA and P57 mRNA were almost unaffected by circTCF4 deficiency (Figure 3A). Furthermore, using the CCK-8 assay, we found that si-circTCF4 treatment significantly enhanced the absorbance of the cells at 24 h, 48 h, and 72 h, respectively (*p* < 0.01) (Figure 3C). Additionally, we employed the EdU assay and found that circTCF deficiency elevated the number of newly formed nuclei by ~30% (Figure 3D,E).

We further constructed an overexpression vector of circTCF4 (PCD5-circTCF4) and transfected it into goat MuSCs. As expected, circTCF4 transcripts significantly increased in PCD5-circTCF4-treated cells (*p* < 0.001), whereas the levels of TCF4 mRNA, Pax7 mRNA, PCNA mRNA, CCND1 mRNA, CDK2 mRNA and P57 mRNA were slightly altered (*p* > 0.05) (Appendix A). Besides, ectopic circTCF4 failed to significantly change the absorbance of the cells at 24 h, 48 h, and 72 h, respectively (*p* > 0.05) (Appendix A) and the number of newly formed nuclei (Appendix A). Combined with the promoting effect caused by circTCF deficiency in cells, we speculate that circTCF4 is outset likely in a saturated state in proliferating MuSCs. These suggest that circTCF4 plays a critical role in the proliferation of MuSCs cells, which is likely independent of the linear transcripts of the TCF4 gene.

### 2.4. CircTCF4 Suppressed Goat MuSCs Differentiation

To investigate the effect of circTCF4 on the differentiation of Goat MuSCs, we transfected the overexpression vector (PCD5-circTCF4) and empty vector (PCD5-cir) into MuSCs-initiated differentiation, respectively. Similar to that presented in proliferating stage, a highly significant increase in circTCF4 levels (*p* < 0.01) (Figure 4A) was accompanied by an insignificant change in TCF4 mRNA (Figure 4B). Notably, ectopic circTCF4 dramatically decreased expressions of myogenic differentiation-related marker genes, such as Myomaker, MyHC, Myomerger, MyoD, and MyoG transcripts (*p* < 0.01) (Figure 4A). To confirm this result, we detected MyoD and MyHC protein using a Western Blot assay and found that they were downregulated by circTCF4 (Figure 4C), as expected. Moreover, using MyHC immunofluorescence staining, we observed the shortened myofibers and the decreased number of MyHC-positive cells caused by PCD5-circTCF4 treatment compared with the control (Figure 4D).

On the contrary, circTCF4 deficiency, which resulted from siRNA specifically targeting circTCF4 (si-circTCF4), failed to change the expression of TCF4 mRNA (*p* > 0.05) (Figure 4F) but significantly enriched Myomaker, MyHC, Myomerger, MyoD, and MyoG mRNA (Figure 4E) as well as MyoD and MyHC protein, compared with the control (Figure 4G). Also, results from the MyHC immunofluorescence staining assay confirmed an increase in the number and length of the MyHC-positive cells in the circTCF4 deficiency group (Figure 4H).

In summary, these imply the inhibitory effect of circTCF4 on the differentiation process of MuSCs.

### 2.5. Signaling Pathways Involved in CircTCF4-Related MuSCs Differentiation

We employed mRNA transcriptome sequencing to systematically screen the downstream genes affected by circTCF4 at the differentiation stage. The mycoplasma-free cells transfected with si-circTCF4 or control were sampled (Appendix A), and their total RNAs were extracted and qualified (Appendix A), followed by PolyA RNA enrichment and sequencing. As expected, siRNA transfection significantly reduced circTCF4 levels (Figure 5A), and expression correlations among six sequencing libraries were above 0.953, roughly validating the success of our experiments (Figure 5B). A total of 5352 mRNAs were dysregulated significantly, of which 2652 mRNAs were up-regulated and 2727 down-regulated (Figure 5C). Additionally, these genes were well clustered according to the treatment (Figure 5D). Based on RNA-seq data, eight randomly selected genes presented similar expression patterns validated via RT-qPCR (Appendix A), reflecting the reliability of mRNAs identified in this study.

Moreover, the 2727 downregulated transcripts caused by deficiency of circTCF4 were enriched in RNA binding, chromatin, and mitochondrion organization (Figure 5E). In contrast, those 2652 up-regulated transcripts were significantly enriched into 23 GO entries (*p*adj < 0.05) and 29 signaling pathways (*p*adj < 0.05). Among them, the well-known myogenic pathways, including the insulin signaling pathway and AMPK signaling pathway, were successfully screened (Figure 5F). In addition, by Gene set enrichment analysis, we found that the differentially expressed genes were also enriched in the insulin signaling pathway and AMPK signaling pathway, which affect MuSCs differentiation (Figure 5G). Subsequently, we constructed protein-protein interaction networks for differential genes in these two signaling pathways, and we found that AKT1, AKT2, MAPK3, and IGF1R proteins associated with differentiation are important nodes (Figure 5H).

### 2.6. CircTCF4 Regulated the Proliferation and Differentiation of MuSCs Independent from AGO2 Binding

CircRNAs regulate their target genes via sponging miRNA, a process closely connected with AGO2 protein. To investigate whether the inhibition of circTCF4 in the proliferation and differentiation process of Goat MuSCs is mediated by miRNAs, using the online miRNA binding site prediction software RNA22 (version 2.0) and RNAhybird, we found that circTCF4 potentially contains miR-671-3p and miR-1343 binding sites. Subsequently, we employed miRanda, TargetScan, and PicTar software (version 1.0) and identified complementary base pairing between the transcript of Pax7 and miR-671-3p, as well as MyoD and miR-1343 (Figure 6A). Unexpectedly, both overexpression and deficiency of circTCF4 failed to significantly alter the expression of miR-671-3p and miR-1343 (Figure 6B). Moreover, we performed AGO2-RNA immunoprecipitation to explore the combination between RNAs and AGO2 protein in the proliferation and differentiation of cells. As a result, the circTCF4 was insignificantly enriched by the AGO2 antibody compared with the control IgG (Figure 6C). These suggest that circTCF4 unlikely functions in myogenesis by competitively binding miRNAs.

CircRNA can initiate translation through RNA methylation modification, which requires the involvement of methyltransferases METTL3 and METTL14, as well as the methylation recognition protein YTHDF family. Interestingly, we analyzed the RNA binding protein motifs on circTCF4 sequences using catRAPID, and we found that circTCF4 has potential motifs for YTHDF1, YTHDF2, YTHDF3, METTL3, and METTL14 (Appendix A). Additionally, circTCF4 potentially has two open reading frames (The purple one spans the back-splicing site of circTCF4), indicating that circRNA-translated peptides, which needed to be verified experimentally, might be one of the functional mechanisms of circTCF4 underneath goat myogenesis (Appendix A).

## 3. Discussion

CircRNAs were thought to be some byproducts of spliceosome-mediated splicing errors or experimental accidents when they were originally found in plant viroids [35,36]. In recent years, with the development of high-tech sequencing technology, a large number of circRNAs have been discovered in mammals. More and more studies have shown that circRNAs play a pivotal role in regulating cell proliferation, apoptosis, and differentiation [37,38]. circRNAs are formed by reverse splicing without 5′ cap and 3′ poly tails; Thus, circRNAs are more stable due to their resistance to RNase [19]. In this study, circTCF4 is formed by 11–14 exons of the *TCF4* gene. It is a typical exon circRNA with strong resistance to RNase R. CircTCF4 has a unique cyclization site and is expressed in muscle tissues at a much higher level than other tissues, such as the LD muscle and gastrocnemius. Interestingly, circTCF4 was expressed at higher levels in the middle and late stages of embryonic development and after birth, which is a critical period for muscle fiber maturation. Moreover, circTCF4 increased continually at differentiation stages. Therefore, we hypothesized that circTCF4 has an important effect on the proliferation and differentiation of MuSCs, and understanding the functional mechanism of circTCF4 could provide new insights into muscle development in goats.

To test our hypothesis, we used overexpression vector and interference fragments to verify whether circTCF4 has a momentous role in MuSCs development. In proliferating MuSCs, deficiency of circTCF4 dramatically elevated the expression of Pax7, a marker gene for MuSCs proliferation, and promoted MuSCs proliferation. Meanwhile, overexpression of circTCF4 had no significant effect on MuSCs proliferation, which is likely that circTCF4 reached functional saturation. In differentiating cells, In the differentiation stage of MuSCs, ectopic circTCF4 or deficiency significantly decreased or increased the expression levels of MyoD, MyoG, Myomaker, Myomerger, and MyHC genes, and myotubes formation. Intriguingly, the interference of circTCF4 did not affect linear TCF4 mRNA. These results provide robot evidence that circTCF4 inhibits myogenic proliferation and differentiation.

It is well-known that several molecular signaling pathways, such as the Wnt-, BMP4-, p38 MAPK-, insulin signaling-, and AMPK signaling pathway, participate in the proliferation and differentiation process of MuSCs by regulating the expression of myogenic regulatory factors [25,39,40,41,42]. To explore the signaling pathways circTCF4-involved in MuSCs differentiation, we transfected si-circTCF4 into differentiating MuSCs and then performed mRNA transcriptome sequencing. A total of 2652 up-regulated mRNAs were screened and enriched in 29 signaling pathways. As expectedly, those classical myogenic pathways, such as insulin signaling pathway, AMPK signaling pathway, forkhead-like transcription factor signaling pathway (FoxO signaling pathway), and so on, were identified. Additionally, we found that the differentially expressed genes were also enriched in the insulin signaling pathway and AMPK signaling pathway. Subsequently, we constructed protein-protein interaction networks, and found that AKT1, AKT2, MAPK3, and IGF1R proteins associated with differentiation are important nodes.

Currently, several molecular mechanisms underpinning circRNAs, such as miRNA sponge (ceRNA mechanism) [25,43], carrier for RNA-binding protein [24], and even circRNA-translated peptides [44], have been discovered in the proliferation and differentiation of MuSCs [45,46]. Among these, the ceRNA mechanism has been ranked first. For example, a classical circRNA CDR1as, containing many miR-7 binding sites, promotes the differentiation of goat MuSCs via elevating the expression of IGF1R, a downstream gene potentially targeted and degraded by miR-7 [25]. Therefore, to explore the molecular mechanism of circTCF4 in goat myogenesis, our priority target is to investigate the miRNA binding sites of circTCF4. Indeed, using online tools, we found two miRNAs, miR-671-3p and miR-1343, showing potential binding on circTCF4, and 3′UTR region of *Pax7* mRNA and *MyoD* mRNA, respectively. Since several studies have shown that circRNAs may change their target binding miRNAs after being overexpressed or disrupted [47,48], we initially examined the expression of miR-671-3p and miR-1343 and found no significant changes caused by circTCF4 alteration. We also noticed that deficiency of circTCF4 promoted Pax7 expression in proliferation cells, overexpression and interference of circTCF4 inhibited and promoted MyoD expression during the differentiation period, which is contradictory to the trend of ceRNA-related miRNA targets. Moreover, we performed AGO2 RIP experiments to investigate the binding potential of miRNAs on circTCF4, and found that AGO2 did not significantly enrich circTCF4. Based on these results, we concluded that circTCF4 did not function by binding miR-671-3p and miR-1343. Since a few circRNAs have been reported to translate into proteins in cap-independent ways [26,49]. For instance, CircZNF609 contains a 753 nt open reading frame and regulates the proliferation and differentiation of C2C12 myoblasts via IRES-mediated translation peptides [44]. Accordingly, we found that circTCF4 potentially has two open reading frames (One spans the back-splicing site of circTCF4), indicating that circRNA-translated peptides, which needed to be verified experimentally, might be one of the functional mechanisms of circTCF4 underneath goat myogenesis.

## 4. Materials and Methods

### 4.1. Ethics Statement

In this study, all the experimental schemes were approved by the Institutional Animal Care and Utilization Committee of Sichuan Agricultural University, under permit No. DKY-2020202011. As well, they were conducted according to the Regulations for the Administration of Affairs Concerning Experimental Animals (Ministry of Science and Technology, Chengdu, China).

### 4.2. Animals and Samples Collection

Animals used here were Jianzhou Big-Eared goats from the Jianyang Dageda farm (Sichuan, China). Generally, all the pregnant ewes (healthy, 2–3 years old) were housed in free stall and fed a standard diet (forage to concentrate ratio, 65:35), twice (06:30–08:30 and 16:00–18:00) per day, and water ad libitum. The goat fetuses and kids (female, n = 3 per stage) at 45, 60, and 105 day of gestation (E45, E60, E105), as well as 3, 60, 150 day postnatal (B3, B60, B150) were randomly chosen, humanely collected by caesarean section, and sacrificed. A total of 11 tissues including five internal organs (heart, liver, lung, spleen, and kidney), five skeletal muscles (longissimus dorsi M., LD; Semimembranosus M.; Semitedinosus M.; Psoas major M.; and Gastrocnemius M.), and Cerebrum were sampled and snap frozen in liquid nitrogen, and stored at −80 °C for later use.

### 4.3. Skeletal Muscle Satellite Cells (MuSCs) Isolation

MuSCs used in this experiment were successfully separated from the LD muscle of neonatal goats, as described previously [25]. Briefly, LD tissue blocks were quickly rinsed with sterile phosphate buffer (PBS, Hyclone) three times, then cut into pieces with medical scissors and digested with 0.2% Pronase (Sigma-Aldrich, St. Louis, MO, USA) at 37 °C for 1 h, and centrifuged at 1500× *g* for 6 min to precipitate cell pellet. The cell pellet was resuspended in Dulbecco’s modified Eagle’s medium (DMEM/high glucose, Hyclone) supplemented with 15% fetal bovine serum (Gibco, Grand Island, NY, USA). MuSCs were obtained by filtrating the cell suspension with a 70-μm mesh sieve and centrifuging at 800× *g* for 5 min. To purify MuSCs, we used the Percoll gradients (90, 40, and 20%) (Sigma-Aldrich) to enrich MuSCs from 40% to 90% of the interface and then immunostained cells with Pax7 antibody (pairing box 7, rabbit anti-PAX7, 1:100 dilution, Boster, Wuhan, China), a key marker of MuSCs. Finally, the qualified Pax7^+^ MuSCs were stored in liquid nitrogen.

### 4.4. MuSCs Culture and Transfection

In principle, MuSCs were seeded in 6-well (~2 × 10^4^ cells per well) or 12-well (~1 × 10^4^ cells per well) culture plates and cultured in a growth medium (GM) consisting of 88% DMEM, 10% FBS (Gibco, Grand Island, NY, USA), and 2% solution of penicillin-streptomycin (Invitrogen, Bohemia, NY, USA) in a 37 °C incubator containing 5% CO_2_. When density reached 80% to 90%, cells were placed in differentiating medium (DM) with 2% FBS and 2% Penicillin and Streptomycin to induce differentiation. DM was replaced every two days.

For the gain and loss function study, cells placed in penicillin- and streptomycin-free DM were transfected with Lipofectamine 2000 (Invitrogen, USA), siRNA (si-circTCF4, siNC), overexpression plasmid (PCD5-circTCF4), or control empty plasmid (PCD5-cir). Six hours after transfection, the medium was shifted to regular DM. The RNA and protein were extracted from cells harvested at 48 h and 72 h post-transfection, respectively. MyHC immunofluorescence assay was performed on the 7th day of differentiation. To avoid the bias caused by mycoplasma, we detected mycoplasma in culture media using PCR.

### 4.5. RNA Extraction and qPCR

Total RNAs were extracted from liquid nitrogen-powdered tissues or cultured cells by using RNAiso Plus reagent (Takara, Dalian, China), according to the manufacturer’s specifications. After roughly qualified degradation and contamination by using 1.5% agarose gel electrophoresis, and concentration by utilizing NanoDrop 2000c Spectrophotometer (Thermo-Fisher Scientific, Waltham, UK). Subsequently, the qualified RNAs (~1 mg) were reverse-transcribed into cDNA for mRNA or miRNA assay using the PrimeScript™ RT reagent Kit with gDNA Eraser or miRNA PrimeScript RT reagent Kit (Takara, Japan) separately. Then we accurately quantified the expression of genes by real-time PCR (qPCR) in a Bio-Rad CFX96 system (Bio-Rad, Shanghai, China) with SYBR Premix Ex Taq TM II (Takara, Dalian, China). More than three samples were collected per treatment, and each was technically tri-replicated in the qPCR assay. We employed 2^−ΔΔCt^ method to scale the relative RNA levels of the target genes with GAPDH (Glyceraldehyde-3-Phosphate Dehydrogenase) as the internal control for mRNA or circRNA, U6 for miRNAs. These primers were detailed in Appendix A.

### 4.6. RNase R Treatment

Total RNAs (5 μg) extracted from the LD muscles of newborn goats were purified with DNase three times. The DNA-free RNA was added to 30 μL reaction reagent with 20 units of RNase R (Epicentre, Madison, WI, USA) or 0 units (Control) and 3 μL 10 × RNase R Reaction Buffer (Epicentre), incubated at 37 °C for 1 h. Then we added phenol-chloroform-isoamyl alcohol (30 μL) to stop the reaction immediately. The solution was centrifuged at 13,000× *g* at 4 °C for 5 min to collect the supernatant. Subsequently, the supernatant was mixed thoroughly with 6 μL Licl (4 M), 1 μL glycogen (Thermo, New York, NY, USA), and 90 μL absolute Ethanol (−20 °C) and incubated at −80 °C for 1 h. Finally, washed twice with 75% ethanol (−20 °C) and centrifuged at 13,000× *g* for 5 min at 4 °C, the precipitated RNA was air-dried and entirely resuspended in 20 μL DEPC water and reverse-transcribed into cDNA by using the PrimeScriptTM RT reagent Kit with gDNA Eraser (Takara, Otsu, Japan) for detecting the remaining transcripts by qPCR.

### 4.7. Plasmids Construct and Interfere RNA Design

To get circTCF4 overexpressing vector (PCD5-circTCF4), we amplified the intact circTCF4 from MuSCs cDNA. We conducted it into PCD5-cir (Geneseed Biotech, Guangzhou, China) using double digestion with *EcoRI* and *BamHI* and T4 DNA ligase (Invitrogen, USA), according to the manufacturer’s guidelines. Subsequently, the correction of overexpressing vector was verified by PCR assay combined with Sanger sequencing.

To provide solid interfering results for circTCF4, we designed small interfering RNA (siRNA, AGACACTCACTCATGCAAA) targeting the back splicing sequence of circTCF4, with nonspecific siRNAs sequences as a negative control. RNAs were synthesized by Ribobio (Guangzhou, China).

### 4.8. Cell Counting Kit-8 (CCK-8)

CCK-8 proliferation kit (Biosharp, Hefei, China) was used for counting cells. MuSCs were treated with 0.25% trypsin and then triturated into individual cells with the addition of DMEM. The cells were resuspended in GM, seeded into a 96-well plate (1000 cells per well), dispersed evenly in 100 μL culture medium, and placed in an incubator at 37 °C with 5% CO_2_ and saturated humidity. A 10 μL CCK-8 reagent was added and gently mixed at 0 h, 24 h, 48 h, and 72 h, respectively. After incubating at 37 °C for 2 h, each well’s optical density (OD) was measured by enzyme-linked immunosorbent assay (450 nm measuring wave). Each group was performed in quintuplicate with a minimum of three independent experiments.

### 4.9. EdU Assay

MuSCs (2 × 10^3^ cells per well) were initially cultured and transfected the same as in the CCK-8 assay. Twelve hours after transfection, primary myoblasts were cultured in GM consisting of 10 μM 5-ethynyl-2′-deoxyuridine (EdU; Beyotime China). Every 24 h, MuSCs were fixed (4% PFA at room temperature for 30 min), permeabilized (0.5% Triton X-100), incubated (1 × Apolloreaction cocktail for 30 min), and then stained (1 × Hoechst 33,342 for 10 min). Finally, we quantified the EdU-stained cells (ratio of EdU^+^ myoblasts to all) using randomly selected fields captured shortly after staining by employing an Olympus IX53 inverted microscope (Tokyo, Japan). Assays were performed at least three times.

### 4.10. Western Blotting (WB) Assay

To confirm the expression profiles of target genes, we quantified their protein levels using a classical WB assay. Firstly, the total proteins from in vitro cultured cells were extracted using a radioimmunoprecipitation assay kit (RIPA) (Beyotime, Shanghai, China) and quantified by the bicinchoninic acid assay kit (BCA) (Beyotime, China). Secondly, the qualified protein samples (~20 μg per sample) were loaded separately in polyacrylamide gel electrophoresis and transferred to Polyvinylidene fluoride (PVDF) membranes (Millipore, Burlington, MA, USA). Thirdly, PVDF membranes containing total proteins were incubated with primary anti-rabbit myogenic differentiation 1 (MyoD) (1:1000) and myosin heavy chain (MyHC) (1:1000) (Abclonal, Wuhan, China) at 4 °C overnight, and secondary antibody IgG (1:2000) (Abclonal, China) for 2 h. Finally, after adding Horse Radish Peroxidase (HRP) (Bio-Rad, Hercules, CA, USA), the protein bands were detected and analyzed using electrochemiluminescence (ECL) (Pierce, Appleton, WI, USA) and Image Software (version 6.0.1) with β-Tubulin (1:1000) (Abclonal, China) as a loading control.

### 4.11. MyHC Immunofluorescence Assay

For MyHC immunofluorescence assay, MuSCs (seeded in 3.5-cm Petri dishes with ~2 × 104 cells per dish) cultured in DM for 7 days were fixed with 4% paraformaldehyde (room temperature, 15 min), washed with 1 mL PBS (3 times), permeabilized with 1 mL 0.5% Triton X-100 (4 °C, 10 min), blocked in 1 mL 2% bovine serum albumin (37 °C, 30 min), incubated with anti-rabbit MyHC (1:150, 4 °C, overnight) (Abclonal, China) and secondary antibodies Cy3_IgG (H + L) (1: 200, Abclonal, China) 37 °C for 2 h sequentially. Finally, 0.05μg/mL DAPI (4′, 6′-diamidino-2-phenylindole; Invitrogen) was added to cells and kept in the dark (37 °C for 10 min). Images were captured using an Olympus IX53 inverted microscope (Tokyo, Japan) and then analyzed using ImageJ software (version 2.0).

### 4.12. RNA-Binding Protein Immunoprecipitation (RIP) Analysis

To evaluate the miRNAs’ sponge potential of circTCF4, we employed the RIP assay using the Magna RIP^TM^ RNA-Binding Protein Immunoprecipitation Kit (Millipore, USA) according to the manufacturer’s protocol. In general, cells were collected on the 3rd day of proliferation and the 5th day of differentiation and lysed using Lysis Buffer. Then, Ago2 immunoprecipitation was performed using an anti-Ago2 antibody (Abcam, Cambridge, UK) with an IgG (Millipore, Burlington, MA, USA) as a negative control. Finally, the immunoprecipitated RNA was isolated, and the abundance of circTCF4 was evaluated by qPCR analysis.

### 4.13. mRNA-Seq and Bioinformatic Analyses

Library preparation and poly(A) selection mRNA-seq was performed at Novogene Company (Beijing, China). In brief, using NEBNext^®^ Ultra^TM^ RNA Library Prep Kit Illumina^®^, polyA RNAs (primarily mRNA) were isolated from total RNA samples (200 ng) extracted from si-circTCF4- or siNC-transfected cells (n = 3 per group), followed by fragmentation and generation of double-stranded cDNA. Libraries were evaluated by Qubit 2.0 Fluorometer, Agilent 2100 bioanalyzer, and RT-qPCR. Then qualified libraries were sequenced in Illumina HiSeq 2500 platform (Illumina, San Diego, CA, USA) with a 2 × 150 bp pair-end.

A total of 264,256,682 double-terminal 150 nt raw reads were obtained from these six libraries, generating 38.13 G data. In addition, their Q20 and Q30 were all above 97% and 93%, respectively (Appendix A). Furthermore, 96.63% to 97.7% of clean reads were quickly and accurately mapped onto the Capra hircus ARS1 reference genome (ftp://ftp.ncbi.nlm.nih.gov/genomes/all/GCF/001/704/415/GCF_001704415.1_ARS1/GCF_001704415.1_ARS1_genomic.fna.gz) using HISAT2, among which as high as 90.64% to 92.19% clean reads were uniquely mapped (Appendix A). To evaluate gene expression, we calculated the read count for each one using fragments per kilobase of transcript sequence per million base pairs sequenced (FPKM) value. Differentially expressed genes (DEGs) between samples were canonically identified by DESeq2 R package vision 1.16.1 (|log2(FoldChange)| > 0 and *p*adj < 0.05).

Transcriptome clustering based on principal component analysis (PCA) indicates that si-circTCF4 or siNC-treated cells are distinctly separated. Function Enrichment Analyses of DEGs, including Gene Ontology (GO) enrichment analysis and KEGG pathway, were implemented by using the DESeq2 with *p*adj < 0.05 (adjusted via Benjamini–Hochberg) and was considered significantly enriched.

### 4.14. Bioinformatic Analysis

The targeting interaction between miRNAs and target genes or circTCF4 was calculated by using Targetscan, RNAhybrid, RNA22, miRanda and PicTar [50,51,52,53]. We constructed protein interaction networks using STRING and Cytoscape [54,55]. Then, we employed catRAPID to predict the motif for RNA protein binding on circTCF4 [56]. Subsequently, the open reading frame of circTCF4 was predicted using circAtlas [57].

### 4.15. Statistical Analysis

Data are means ± standard error, with at least three biological replicates. The grouped two-tailed *t*-test was used for comparing two groups using GraphPad Prism 9.0, * *p* < 0.05, ** *p* < 0.01 and *** *p* < 0.001 were considered statistically significant.

## 5. Conclusions

We identified a novel muscular circRNA named circTCF4, a circular transcript of the TCF4 gene. Independence of its linear transcript, circTCF4 presents an inhibitory effect on the proliferation and differentiation of MuSCs, involving multiple muscle-differentiation-related signaling pathways. Meanwhile, circTCF4 is unlikely to function through the AGO2-mediated ceRNA mechanism. These results further the molecular mechanism of proliferation and differentiation of MuSCs in goats.

## Figures and Tables

**Figure 1 ijms-23-12868-f001:**
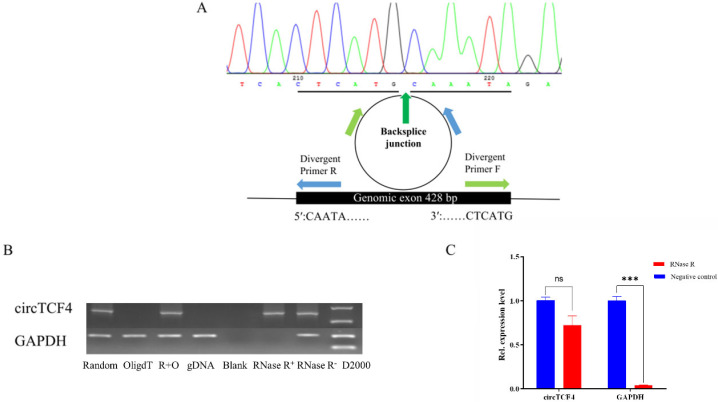
Characterization of Goat circTCF4. (**A**), Sequencing results of circTCF4 splicing junction. (**B**), PCR amplification results of circTCF4 in cDNA obtained by Random primer reverse transcription, OligdT primer reverse transcription, Random + OligdT primer reverse transcription, gDNA (genome DNA), and blank control, and RNase R treated group, respectively. (**C**), circTCF4 and GAPDH abundance in RNase R digested samples. Data are means ± standard error of the mean of at least three biological replicates, *** *p* < 0.001.

**Figure 2 ijms-23-12868-f002:**
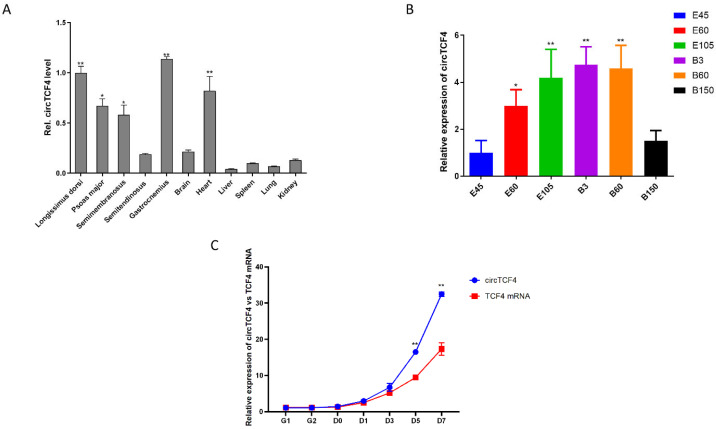
Expression pattern of goat circTCF4. (**A**), Relative abundance of circTCF4 in different tissues of Goats. (**B**), Changes of circTCF4 levels during the development of goat longissimus dorsi muscle. (**C**), Changes of circTCF4 and *TCF4* mRNA levels during proliferation and differentiation of MuSCs. Data are means ± standard error of the mean of at least three biological replicates, * *p* < 0.05 and ** *p* < 0.01.

**Figure 3 ijms-23-12868-f003:**
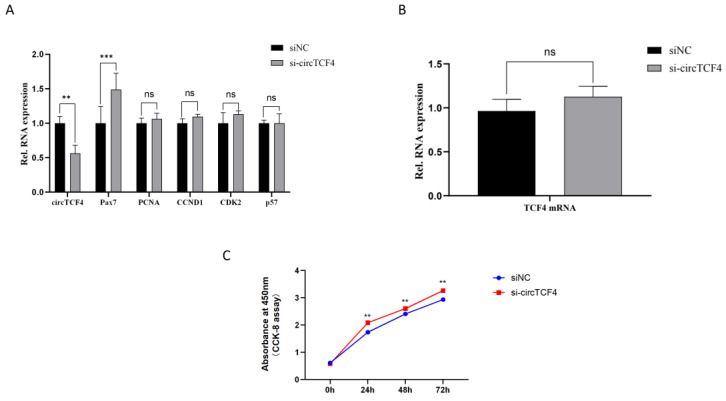
Effect of circTCF4 deficiency on the proliferation of MuSCs in goats. (**A**) Changes in circTCF4, *Pax7* mRNA, *PCNA* mRNA, *CCND1* mRNA, *CDK2* mRNA, and *P57* mRNA expression altered by interfering with circTCF4. (**B**) Changes in *TCF4* mRNA expression. (**C**) The absorbance of cells at 450 nm. (**D**) The number of EdU stained nuclei in cells. (**E**) Statistical results of EdU positive cells. Data are means ± standard error from at least three biological replicates, ** *p* < 0.01 and *** *p* < 0.001.

**Figure 4 ijms-23-12868-f004:**
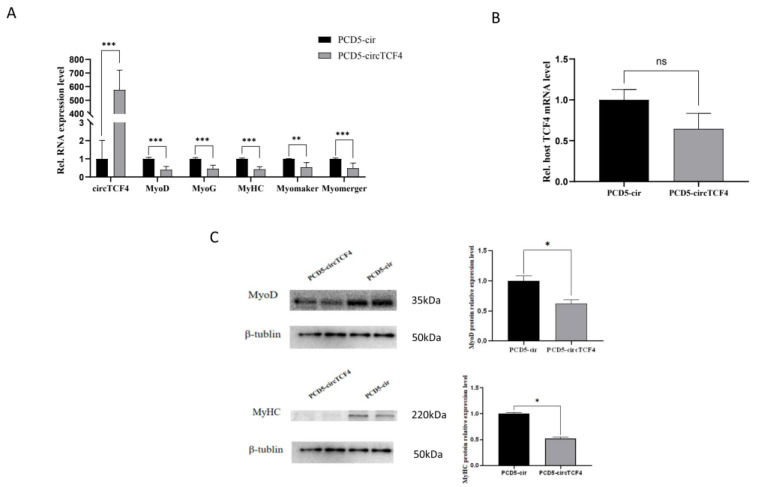
Effect of interference with circTCF4 on the differentiation of MuSCs. (**A**) Expression of circTCF4 and myogenic differentiation marker genes (Myomaker, MyHC, Myomerger, MyoD, and MyoG) after overexpression of circTCF4. (**B**) levels *TCF4* mRNA after circTCF4 overexpression. (**C**) MyoD and MyHC protein levels after overexpression of circTCF4. (**D**), MyHC-stained myofibers after overexpression of circTCF4 and statistical results of MyHC-positive cells and myogenic index after overexpression. (**E**) levels of circTCF4 and MuSCs differentiation marker genes (Myomaker, MyHC, Myomerger, MyoD and MyoG) transcripts altered by circTCF4 deficiency. (**F**) *TCF4* mRNA expression after circTCF4 interference. (**G**) MyoD and MyHC protein levels changed by circTCF4 interference. (**H**). MyHC-stained myofibers affected by circTCF4 deficiency and statistical results of MyHC positive cells and myogenic index after interference with circTCF4. Data are means ± standard error from at least three biological replicates, * *p* < 0.05, ** *p* < 0.01, and *** *p* < 0.001.

**Figure 5 ijms-23-12868-f005:**
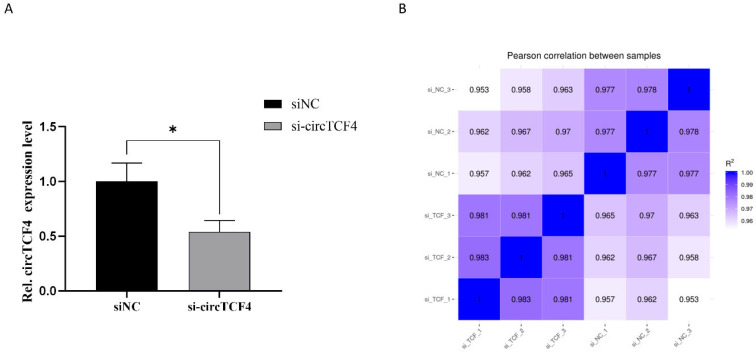
Signaling pathways involved in circTCF4-related differentiation of MuSCs. (**A**) CircTCF4 expression after circTCF4 interference. (**B**) Heat map of correlation between samples. (**C**) Volcano map of differentially expressed genes. (**D**) Heat map of differentially expressed genes. (**E**) GO enrichment of down-regulated mRNA host genes. (**F**) KEGG signaling pathway of up-regulated mRNA host genes. (**G**) Gene set enrichment analysis of differentially expressed genes. (**H**) Protein-protein interaction networks of differentially expressed genes involved in the MuSCs differentiation. Data are means ± standard error from at least three biological replicates, * *p* < 0.05.

**Figure 6 ijms-23-12868-f006:**
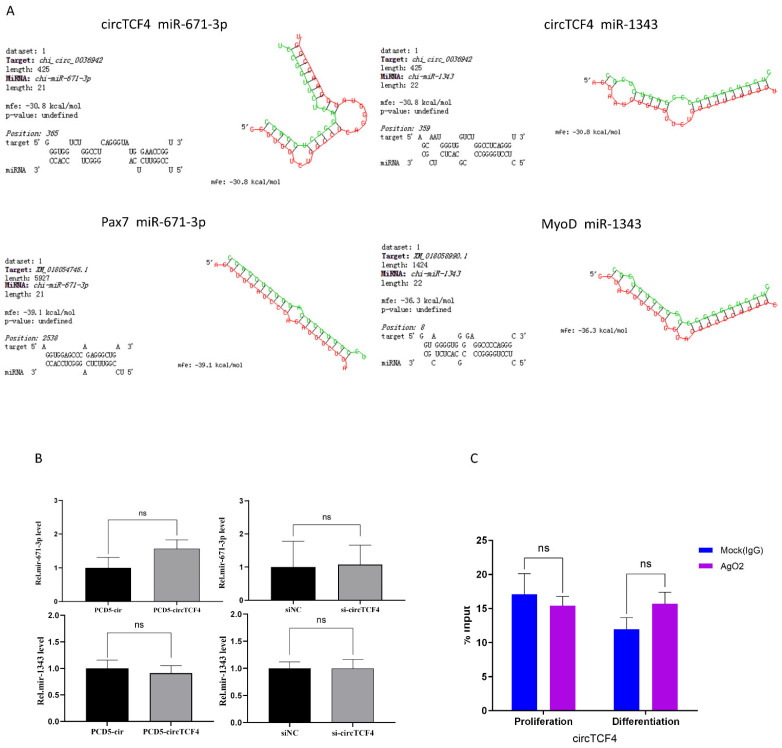
CircTCF4 regulates the proliferation and differentiation of MuSCs independence of miRNAs. (**A**) Predicted binding sites between circTCF4 and mir-671-3p and mir-1343 and downstream target genes. (**B**) Effects of circTCF4 on miR-671-3p and miR-1343 expression. (**C**) circTCF4 enriched in AgO2-RIP. Data are means ± standard error of the mean of at least three biological replicates, ns means insignificant.

## Data Availability

The datasets generated/analyzed during the current study are available. The raw sequencing data are available through the NCBI data accession number PRJNA882586.

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
