# Peer review of "CircTCF4 Suppresses Proliferation and Differentiation of Goat Skeletal Muscle Satellite Cells Independent from AGO2 Binding"

_ijms, 2022, doi:10.3390/ijms232112868_

Round 1

Reviewer 1 Report

In the manuscript “Suppression of circTCF4 on the proliferation and differentiation of goat skeletal muscle satellite cells” by Shuailong Zheng et al., the authors identify a novel muscular circular RNA, circTCF4, with an inhibitory effect on the proliferation and differentiation of goat skeletal muscle satellite cells (MuSCs). They show that circTCF4 is unlikely to function through the AGO2-mediated ceRNA mechanism. The results reveal a novel molecular mechanism in the control of the proliferation and differentiation processes in MuSCs in goat.

Major points

1) Figure 4: the myogenic index, i.e. the percentages of nuclei inside the myotubes (3 or more nuclei) with respect to total nuclei after May-Grünwald/Giemsa staining, should be also performed.

Minor points

1) The title should be modified into a more appropriated form.

2) The Abstract and the text should be extensively edited for English language and style.

3) Abbreviations should be listed in alphabetical order.

Author Response

Summary: In the manuscript “Suppression of circTCF4 on the proliferation and differentiation of goat skeletal muscle satellite cells” by Shuailong Zheng et al., the authors identify a novel muscular circular RNA, circTCF4, with an inhibitory effect on the proliferation and differentiation of goat skeletal muscle satellite cells (MuSCs). They show that circTCF4 is unlikely to function through the AGO2-mediated ceRNA mechanism. The results reveal a novel molecular mechanism in the control of the proliferation and differentiation processes in MuSCs in goat.

We greatly appreciate your encouraging comments and thank you for your insightful suggestions, these are great questions and we share with your understandings as below.

Major points

  • Figure 4: the myogenic index, i.e. the percentages of nuclei inside the myotubes (3 or more nuclei) with respect to total nuclei after May-Grünwald/Giemsa staining, should be also performed.

Response: Thank you for your comments. According to your suggestion, we calculated the myogenic index and added it to Figure 4D and Figure 4H.

Minor points

  • The title should be modified into a more appropriated form.

Response: Thank you for your comments. According to your suggestion, we revised the title to CircTCF4 suppresses proliferation and differentiation of goat skeletal muscle satellite cells independent from AGO2 binding. Thank you again for your advice.

2) The Abstract and the text should be extensively edited for English language and style.

Response: Thank you for your valuable and thoughtful comments. We have carefully checked and improved the English writing in the revised manuscript.

3) Abbreviations should be listed in alphabetical order.

Response: Thank you for your comments. We have sorted the Abbreviations alphabetically by initials. 

Reviewer 2 Report

The authors conducted a good research on Suppression of circTCF4 on the proliferation and differentiation of goat skeletal muscle satellite cells. However it needs improvement before publication, the main comments below.

Introduction section:

Explain the purpose of the study and then move on to what needs to be done and what problems need to be solved in the last paragraph.

L70: References need to be added at 105 days when muscle fibers begin to form.

Result section:

L92: In Figure 1C, circTCF4, GAPDH treatment group and control group are not significant? Please add significance or not.

L114: In Figure 2C, please explain the abscissa. Why is it G1, G2?

P<0.05 in full text, P should be in italics.

P145: What do "**" and "***" stand for?

L137-140: Please explain what this passage means, or how you speculate that CircTCF4 may initially be saturated in proliferating MuSCs and then draw other conclusions.

Figure 4C: In the second WB, please put in the original image. I think it's a little inappropriate.

L214-215, L228-230: Do not include references in the results.

Figure 6A is not clear, please change a clear picture.

Discussion section:

The discussion section needs an overhaul. The author has done a lot of work, and the results of the content are also very rich, but the discussion is more like a summary.

L252-263: This part belongs to the literature introduction and should be placed in the preface. It is not suitable for description in the discussion. Please reduce or remove it.

Most of the discussion content is an introduction to the literature, and the results of this trial are not discussed. Please improve it.

Material method

L302: Please add ethics approval number.

L307: Describe in detail the sampling process of longissimus dorsi muscle. How the animals were slaughtered, how long it took for the samples to be collected, whether they need to be processed, etc. In addition, please provide the gender of the sheep in different stages and feeding conditions.

L310: Please explain why you chose these periods and what is the significance of them? Is the choice based? Or any random time period.

L331: Incorrect punctuation after 5% CO2.

L342-344: Please supplement the detection method for RNA concentration.

L350: I see two methods in text. Which method was used for the analysis in this study?

Author Response

Summary: The authors conducted a good research on Suppression of circTCF4 on the proliferation and differentiation of goat skeletal muscle satellite cells. However it needs improvement before publication, the main comments below.

We greatly appreciate your encouraging comments and thank you for your insightful suggestions, these are great questions and we share with your understandings as below.

Introduction section:

Explain the purpose of the study and then move on to what needs to be done and what problems need to be solved in the last paragraph.

Response: Thank you for your comments.

Based on the circular RNA-specific transcriptome sequencing results of the goat longissimus dorsi muscle at different stages, we screened for circTCF4, a circRNA molecule with spatial and temporal expression specificity. circTCF4 was consistently highly expressed at 105 d (when the muscle fibers started to form) and 3 d and 60 d after birth, suggesting that it may be involved in regulating muscle growth and development[33,34]. Combined with the results of previous experiments, we selected circTCF4 as a candidate circRNA, and focused on the core question of whether circTCF4 regulates myoblast proliferation and differentiation in this experiment. Our results showed that circTCF4 can efficiently inhibit proliferation and differentiation of MuSCs. Further examination revealed that circTCF4 functions independently from AGO2 binding. These will shed light on the current understanding of circRNA function during muscle progression.

L70: References need to be added at 105 days when muscle fibers begin to form.

Response: Thank you for your comments. According to your suggestion, we have added references in the text (PMID: 27550073 and PMID: 25031653). Thanks again for your suggestion.

Result section:

L92: In Figure 1C, circTCF4, GAPDH treatment group and control group are not significant? Please add significance or not.

Response: Thank you for your comments. We have added significance (Figure 1C).

L114: In Figure 2C, please explain the abscissa. Why is it G1, G2?

Response: Thank you for your advice. G1 represents the first day of the proliferative phase and G2 represents the second day.

P<0.05 in full text, P should be in italics.

Response: Thank you for your comments. We have reviewed the full text carefully and revised it.

P145: What do "**" and "***" stand for?

Response: Thank you for your guidance. "**" stand for p<0.01, "***" stand for p<0.001.

L137-140: Please explain what this passage means, or how you speculate that CircTCF4 may initially be saturated in proliferating MuSCs and then draw other conclusions.

Response: Thank you for your comments. We found that interference with circTCF4 promoted the expression of the proliferation marker gene Pax7 and cell numbers, implying circTCF4 affects MuSCs proliferation. While overexpression of circTCF4 caused no significant change in the expression of the proliferation marker gene and cell number, therefore, we speculated that levels of endogenous circTCF4 was likely saturated.

Figure 4C: In the second WB, please put in the original image. I think it's a little inappropriate.

Response: Thank you for your comments. We have replaced it with the original image.

L214-215, L228-230: Do not include references in the results.

Response: Thanks for your suggestion. We have removed the references in the results section.

Figure 6A is not clear, please change a clear picture.

Response: Thank you for your valuable advice. We replaced the picture with a clearer one.

Discussion section:

The discussion section needs an overhaul. The author has done a lot of work, and the results of the content are also very rich, but the discussion is more like a summary.

Response: According to your suggestion, we have rewritten the discussion section to improve the readability of the article. Thank you for your advice.     

  1. Discussion

CircRNAs were thought to be some byproducts of spliceosome-mediated splicing errors or experimental accidents when they were originally found in plant viroids[35,36]. In recent years, with the development of high-tech sequencing technology, a large number of circRNAs have been discovered in mammals. More and more studies have shown that circRNAs play a pivotal role in regulating cell proliferation, apoptosis, and differentiation[37,38]. circRNAs are formed by reverse splicing without 5′ cap and 3′ poly tails; Thus, circRNAs are more stable due to their resistance to RNase[19]. In this study, circTCF4 is formed by 11-14 exons of the TCF4 gene. It is a typical exon circRNA with strong resistance to RNase R. CircTCF4 has a unique cyclization site and is expressed in muscle tissues at a much higher level than other tissues, such as the LD muscle and gastrocnemius. Interestingly, circTCF4 was expressed at higher levels in the middle and late stages of embryonic development and after birth, which is a critical period for muscle fiber maturation. Moreover, circTCF4 increased continually at differentiation stages. Therefore, we hypothesized that circTCF4 has an important effect on the proliferation and differentiation of MuSCs, and understanding the functional mechanism of circTCF4 could provide new insights into muscle development in goats. 

To test our hypothesis, we used overexpression vector and interference fragments to verify whether circTCF4 has a momentous role in MuSCs development. In proliferating MuSCs, deficiency of circTCF4 dramatically elevated the expression of Pax7, a marker gene for MuSCs proliferation, and promoted MuSCs proliferation. Meanwhile, overexpression of circTCF4 had no significant effect on MuSCs proliferation, which is likely that circTCF4 reached functional saturation. In differentiating cells, In the differentiation stage of MuSCs, ectopic circTCF4 or deficiency significantly decreased or increased the expression levels of MyoD, MyoG, Myomaker, Myomerger, and MyHC genes, and myotubes formation. Intriguingly, the interference of circTCF4 did not affect linear TCF4 mRNA. These results provide robot evidence that circTCF4 inhibits myogenic proliferation and differentiation.

It is well-known that several molecular signaling pathways, such as the Wnt-, BMP4-, p38 MAPK-, insulin signaling-, and AMPK signaling pathway, participate in the proliferation and differentiation process of MuSCs by regulating the expression of myogenic regulatory factors[25,39-42]. To explore the signaling pathways circTCF4-involved in MuSCs differentiation, we transfected si-circTCF4 into differentiating MuSCs and then performed mRNA transcriptome sequencing. A total of 2652 up-regulated mRNAs were screened and enriched in 29 signaling pathways. As expectedly, those classical myogenic pathways, such as insulin signaling pathway, AMPK signaling pathway, forkhead-like transcription factor signaling pathway (FoxO signaling pathway), and so on, were identified. Additionally, we found that the differentially expressed genes were also enriched in the insulin signaling pathway and AMPK signaling pathway. Subsequently, we constructed protein-protein interaction networks, and found that AKT1, AKT2, MAPK3, and IGF1R proteins associated with differentiation are important nodes.

Currently, several molecular mechanisms underpinning circRNAs, such as miRNA sponge (ceRNA mechanism)[25,43], carrier for RNA-binding protein[24], and even circRNA-translated peptides[44], have been discovered in the proliferation and differentiation of MuSCs[45,46]. Among these, the ceRNA mechanism has been ranked first. For example, a classical circRNA CDR1as, containing many miR-7 binding sites, promotes the differentiation of goat MuSCs via elevating the expression of IGF1R, a downstream gene potentially targeted and degraded by miR-7 [25]. Therefore, to explore the molecular mechanism of circTCF4 in goat myogenesis, our priority target is to investigate the miRNA binding sites of circTCF4. Indeed, using online tools, we found two miRNAs, miR-671-3p and miR-1343, showing potential binding on circTCF4, and 3'UTR region of Pax7 mRNA and MyoD mRNA, respectively. Since several studies have shown that circRNAs may change their target binding miRNAs after being overexpressed or disrupted[47,48], we initially examined the expression of miR-671-3p and miR-1343 and found no significant changes caused by circTCF4 alteration. We also noticed that deficiency of circTCF4 promoted Pax7 expression in proliferation cells, overexpression and interference of circTCF4 inhibited and promoted MyoD expression during the differentiation period, which is contradictory to the trend of ceRNA-related miRNA targets. Moreover, we performed AGO2 RIP experiments to investigate the binding potential of miRNAs on circTCF4, and found that AGO2 did not significantly enrich circTCF4. Based on these results, we concluded that circTCF4 did not function by binding miR-671-3p and miR-1343. Since a few circRNAs have been reported to translate into proteins in cap-independent ways[26,49]. For instance, CircZNF609 contains a 753 nt open reading frame and regulates the proliferation and differentiation of C2C12 myoblasts via IRES-mediated translation peptides[44]. Accordingly, we found that circTCF4 potentially has two open reading frames (One spans the back-splicing site of circTCF4), indicating that circRNA-translated peptides, which needed to be verified experimentally, might be one of the functional mechanisms of circTCF4 underneath goat myogenesis.

L252-263: This part belongs to the literature introduction and should be placed in the preface. It is not suitable for description in the discussion. Please reduce or remove it.

Response: Thank you for your comments. According to your suggestion, we have removed L252-263 in the discussion section.

Most of the discussion content is an introduction to the literature, and the results of this trial are not discussed. Please improve it.

Response: According to your suggestion, we have discussed the results of the article. Thank you again for your suggestion.

Material method

L302: Please add ethics approval number.

Response: Thank you for your advice. All research involving animals was conducted according to the regulation proposed by the Institutional Animal Care and Use Committee at Sichuan Agricultural University, under permit No. DKY-2020202011. We have added the ethics approval number in the article.

L307: Describe in detail the sampling process of longissimus dorsi muscle. How the animals were slaughtered, how long it took for the samples to be collected, whether they need to be processed, etc. In addition, please provide the gender of the sheep in different stages and feeding conditions.

Response: Thank you for your comments.

 Animals used here were Jianzhou Big-Eared goats from the Jianyang Dageda farm (Sichuan, China). Generally, all the pregnant ewes (healthy, 2–3 years old) were housed in free stall and fed a standard diet (forage to concentrate ratio, 65:35), twice (06:30–08:30 and 16:00–18:00) per day, and water ad libitum. The goat fetuses and kids (female, n = 3 per stage) at 45, 60, and 105 day of gestation (E45, E60, E105), as well as 3, 60, 150 day postnatal (B3, B60, B150) were randomly chosen, humanely collected by caesarean section, and sacrificed. A total of 11 tissues including five internal organs (heart, liver, lung, spleen, and kidney), five skeletal muscles (longissimus dorsi M., LD; Semimembranosus M.; Semitedinosus M.; Psoas major M.; and Gastrocnemius M.), and Cerebrum were sampled and snap frozen in liquid nitrogen, and stored at -80℃ for later use.

L310: Please explain why you chose these periods and what is the significance of them? Is the choice based? Or any random time period.

Response: Thank you for your comments.

     Morphological studies in our laboratory on the longissimus dorsi M. of female fetuses at different developmental stages of Jianzhou Big-Eared goats, i.e. 45 d (E45), 60 d (E60), 75 d (E75), 90 d (E90), 105 d (E105), 120 d (E120), 135 d (E135) and 3 d after birth (B3), revealed that in the early embryonic development (E45 and E60), mature muscle fibers were not yet formed. and more mature muscle fibers appeared only in the middle and late stages of development (E90 and E105).

   Meanwhile, in terms of growth rate, the difference in growth rate of E45-E60, E60-E105 and E105-B3 reached a significant level, and combined with the morphological observations of muscle fibers, four periods of early fetal development, E45 and E60, middle and late E105, and just after birth B3, were selected for this study. In addition, Jianzhou Big-Eared goats were weaned at 60 d after birth, while at 150 d after birth Jianzhou Big-Eared goats reached the first stage of estrus.

In conclusion, the longissimus dorsi M. at 45d, 60d, 105d and 3d, 60d and 150d after birth were selected for circRNA sequencing studies.

L331: Incorrect punctuation after 5% CO2.

Response: Thank you for your comments. We have corrected this in the article.

L342-344: Please supplement the detection method for RNA concentration.

Response: Thank you for your comments.

Total RNAs were extracted from liquid nitrogen-powdered tissues or cultured cells by using RNAiso Plus reagent (TaKaRa Bio Inc., Japan), according to the manufacturer's specification. After roughly qualified degradation and contamination by using 1.5% agarose gels electrophoresisas, and concentration by utilizing NanoDrop 2000c Spectrophotometer (Thermo-Fisher Scientific, Waltham, MA).

L350: I see two methods in text. Which method was used for the analysis in this study?

Response: Thank you for your comments.

We have corrected this in the article,We employed 2△△Ct  method to scale the relative RNA levels of the target genes with GAPDH (Glyceraldehyde-3-Phosphate Dehydrogenase) as the internal control for mRNA or circRNA, U6 for miRNAs. 

Reviewer 3 Report

The authors investigated the effect of circTCF4 on the proliferation and differentiation of caprine skeletal muscle satellite cells, and also explored the molecular mechanism involved initially. This being a less explored area that could provide relevant insights into the role of circRNA in goat muscle development. However, some aspects of the manuscript need to be improved:

1.      The significance test of difference in Figure 2 should be performed.

2.      The authors described that a strong positive correlation was found in expression between circTCF4 and TCF4 mRNA (lines 111-113, Figure 2C). However, further experiments suggested that the expression of TCF4 mRNA was not changed when the expression of circTCF4 was suppressed (Figure 3B), please explain the contradictory result?

3.      There are many MyHC negative cells in Figure 4D and H. What are these cells? Non-muscle cells or undifferentiated muscle cells?

4.      The authors selected GAPDH gene as the internal control to correct expression of mRNA in qPCR assay. However, in WB assay, β-Tubulin was selected as an internal control. What are the selection criteria for these internal controls?     

5.      A sponge mechanism related to circRNA does not correspond to a change of miRNA expression: miRNA expression remains the same but miRNA can no longer inhibit its mRNA targets. Consequently, please correct the sentence mentioning that circTCF4 failed to significantly alter the expression of miR-671-3p and miR-1343 in lines 221-222. The authors should investigate the effect of circTCF4 on the expression of miRNA targets (PAX7 and MyoD), rather than the expression of miRNAs.   

6.      In proliferation analysis, MTT assay and FACS analysis are important methods to evaluate the cell viability and cycle, why the two assays were not performed in this study?

7.      In the proliferation analysis, the authors selected PAX7 and PCNA as the proliferation marker genes, what is the selection criteria? Why didn't the authors select other marker genes such as CCND1, p27, p57, CDK2 and CDK4?  

8.      In myogenic differentiation assay, the authors selected five myogenic differentiation marker genes to perform qPCR analysis. However, only two of these myogenic regulators (MyoD and MyHC) were selected to perform WB assay, please explain why?

Author Response

Summary:  The authors investigated the effect of circTCF4 on the proliferation and differentiation of caprine skeletal muscle satellite cells, and also explored the molecular mechanism involved initially. This being a less explored area that could provide relevant insights into the role of circRNA in goat muscle development. However, some aspects of the manuscript need to be improved:

We greatly appreciate your encouraging comments and thank you for your insightful suggestions, these are great questions and we share with your understandings as below.

  1. The significance test of difference in Figure 2 should be performed.

Response: Thank you very much for your critical review. We analyzed the data and presented significance of differences in Figure 2.

  1. The authors described that a strong positive correlation was found in expression between circTCF4 and TCF4 mRNA (lines 111-113, Figure 2C). However, further experiments suggested that the expression of TCF4 mRNA was not changed when the expression of circTCF4 was suppressed (Figure 3B), please explain the contradictory result?

Response: Thank you for your valuable and thoughtful comments.

We found the expression of circTCF4 and TCF4 mRNA were strong positive in proliferation and differentiation MuSCs (Figure 2C), suggesting a common process regulates transcripts of circular and linear TCF4. While levels of TCF4 mRNA were unaffected by interference of circTCF4, implying that circTCF4 could not regulate TCF4 mRNA, which excluded the effect of TCF4 mRNA on the proliferation and differentiation caused by circTCF4 (Figure 3B).

  1. There are many MyHC negative cells in Figure 4D and H. What are these cells? Non-muscle cells or undifferentiated muscle cells?

Response: Thank you for your comments. The cells used in this study were satellite cells of skeletal muscle of Jianzhou Big-Eared goats purified and frozen in the laboratory. These MyHC-negative cells in Figure 4D and H are undifferentiated muscle cells.

  1. The authors selected GAPDH gene as the internal control to correct expression of mRNA in qPCR assay. However, in WB assay, β-Tubulin was selected as an internal control. What are the selection criteria for these internal controls?    

Response: Thank you for your comments. Both GAPDH and β-Tubulin are internal reference genes. Since goat are non-model animals, the antibody specificity of GAPDH is very poor and the antibody property of β-Tubulin is better. Therefore, we used GAPDH in qPCR assay, and β-Tubulin in WB assay.

  1. A sponge mechanism related to circRNA does not correspond to a change of miRNA expression: miRNA expression remains the same but miRNA can no longer inhibit its mRNA targets. Consequently, please correct the sentence mentioning that circTCF4 failed to significantly alter the expression of miR-671-3p and miR-1343 in lines 221-222. The authors should investigate the effect of circTCF4 on the expression of miRNA targets (PAX7 and MyoD), rather than the expression of miRNAs.   

Response: We totally agreed that “A sponge mechanism related to circRNA does not correspond to a change of miRNA expression”. Since several studies have shown that circRNAs may change their target binding miRNAs after being overexpressed or disrupted (PMID: 35234473 and PMID: 33542215), we initially examined the expression of miR-671-3p and miR-1343 and found no significant changes caused by circTCF4 alteration. We also noticed that deficiency of circTCF4 promoted Pax7 expression in proliferation cells (Figure 3A), overexpression and interference of circTCF4 inhibited and promoted MyoD expression during the differentiation period (Figure 4A 4E), which is contradictory to the trend of ceRNA-related miRNA targets. Moreover, we performed AGO2 RIP experiments to investigate the binding potential of miRNAs on circTCF4, and found that AGO2 did not significantly enrich circTCF4 (Figure 6C). Based on these results, we concluded that circTCF4 did not function by binding miR-671-3p and miR-1343.

  1. In proliferation analysis, MTT assay and FACS analysis are important methods to evaluate the cell viability and cycle, why the two assays were not performed in this study?

Response: Currently, there are several methods, such as EdU, CCK-8, MTT assay, and FACS, for evaluating cell proliferation, cell viability, or cycle. In our study, we tried to explore the function of circTCF4 on cell proliferation, and EdU and CCK-8 are important methods extensively used for detecting cell numbers. We believe it will be interesting to investigate the function of circTCF4 in cell viability and cycle in the future. Thank you so much for your valuable suggestion.

  1. In the proliferation analysis, the authors selected PAX7 and PCNA as the proliferation marker genes, what is the selection criteria? Why didn't the authors select other marker genes such as CCND1,p27, p57, CDK2 and CDK4?

Response: Pax7 and PCNA are usually chosen as myogenic proliferation markers. According to your suggestion, we further explored the levels of CCND1, p57 and CDK2 during the proliferation period, and found that they were unaffected by overexpression and interference with circTCF4, similar to PCNA (Figure 3A and Supplementary Figure 1A). Thank you again for your suggestion.

  1. In myogenic differentiation assay, the authors selected five myogenic differentiation marker genes to perform qPCR analysis. However, only two of these myogenic regulators (MyoD and MyHC) were selected to perform WB assay, please explain why?

Response: Thank you for your comments. Five myogenic differentiation marker genes, including MyoD, MyoG, MyHC, Myomaker, and Myomerger, were quantified in qPCR, and their levels were halved or doubled by overexpression or deficiency of circTCF4. To further confirm this change, we deliberately chose MyoD and MyHC, markers for early and late differentiation, respectively, to detect their protein levels using WB assay. Their protein levels were consistent with mRNAs. Though it would be perfect if we detected all five proteins, we believe our results reflected the effect of circTCF4 on myogenic differentiation.

Reviewer 4 Report

In this manuscript, the authors attempted to investigate the potential role of circTCF4 in proliferation and differentiation of mammalian skeletal muscle satellite cells (MuSCs). Based on RNA-Seq data, they screened for an exonic circTCF4, which specifically expressed during the development of the longest dorsal muscle in goats. Subsequently, the circular structure and whole sequence of circTCF4 were verified using Sanger sequencing. Besides, circTCF4 was spatiotemporally expressed in multiple tissues from goats but strikingly enriched in muscles. Furthermore, circTCF4 suppressed MuSCs proliferation and differentiation, independent of AGO2 binding. Finally, they conducted Poly(A) RNA-Seq using cells treated with small interfering RNA targeting circTCF4 and found that circTCF4 would affect multiple signaling pathways, including insulin signaling pathway and AMPK signaling pathway related to muscle differentiation.

In general, the manuscript is very well organized and written with a high degree of clarity. However, there are a few minor details that need to be revised. Specific comments are shown below.

1.  Figure 1C needs to mark significance

2.  Figure 4C and Figure 4G need to label the size of the protein molecular weight

3.  Using bioinformatics predictions, the authors found that circTCF4 may function through translation peptides. It is suggested to add relevant statements in the discussion section.

4.  In order to enable readers to quickly understand the content of this research, it is suggested to add a graphic abstract about this paper.

Author Response

Summary:  In this manuscript, the authors attempted to investigate the potential role of circTCF4 in proliferation and differentiation of mammalian skeletal muscle satellite cells (MuSCs). Based on RNA-Seq data, they screened for an exonic circTCF4, which specifically expressed during the development of the longest dorsal muscle in goats. Subsequently, the circular structure and whole sequence of circTCF4 were verified using Sanger sequencing. Besides, circTCF4 was spatiotemporally expressed in multiple tissues from goats but strikingly enriched in muscles. Furthermore, circTCF4 suppressed MuSCs proliferation and differentiation, independent of AGO2 binding. Finally, they conducted Poly(A) RNA-Seq using cells treated with small interfering RNA targeting circTCF4 and found that circTCF4 would affect multiple signaling pathways, including insulin signaling pathway and AMPK signaling pathway related to muscle differentiation.

 In general, the manuscript is very well organized and written with a high degree of clarity. However, there are a few minor details that need to be revised. Specific comments are shown below.

We greatly appreciate your encouraging comments and thank you for your insightful suggestions, these are great questions and we share with your understandings as below.

  1. Figure 1C needs to mark significance

Response: Thank you very much for your critical review. We marked significance in Figure 1C as suggested.

  1. Figure 4C and Figure 4G need to label the size of the protein molecular weight

Response: Thank you for your comments. We have labeled the size of the protein molecular weight in Figure 4C and Figure 4G.

  1. Using bioinformatics predictions, the authors found that circTCF4 may function through

translation peptides. It is suggested to add relevant statements in the discussion section.

Response: Thank you for your valuable and thoughtful comments. According to your suggestion, we have added relevant statements in the discussion section.

  1. In order to enable readers to quickly understand the content of this research, it is suggested to add a graphic abstract about this paper.

Response: Thank you for your comments. According to your suggestion, we added a graphic abstract in this paper.

Round 2

Reviewer 2 Report

No